# Rabies Prophylactic and Treatment Options: An In Vitro Study of siRNA- and Aptamer-Based Therapeutics

**DOI:** 10.3390/v13050881

**Published:** 2021-05-11

**Authors:** Terence Peter Scott, Louis Hendrik Nel

**Affiliations:** Department of Biochemistry, Genetics and Microbiology, University of Pretoria, Pretoria 0002, South Africa; Terence.scott@rabiesalliance.org

**Keywords:** aptamer, siRNA, rabies, post-exposure prophylaxis, treatment, rabies immunoglobulin, lyssavirus

## Abstract

If the goal of eliminating dog-mediated human rabies by 2030 is to be achieved, effective mass dog vaccination needs to be complemented by effective prophylaxis for individuals exposed to rabies. Aptamers and short-interfering RNAs (siRNAs) have been successful in therapeutics, but few studies have investigated their potential as rabies therapeutics. In this study, siRNAs and aptamers—using a novel selection method—were developed and tested against rabies virus (RABV) in a post-infection (p.i.) scenario. Multiple means of delivery were tested for siRNAs, including the use of Lipofectamine and conjugation with the developed aptamers. One siRNA (N53) resulted in an 80.13% reduction in viral RNA, while aptamer UPRET 2.03 demonstrated a 61.3% reduction when used alone at 2 h p.i. At 24 h p.i., chimera UPRET 2.03-N8 (aptamer-siRNA) resulted in a 36.5% inhibition of viral replication. To our knowledge, this is the first study using siRNAs or aptamers that (1) demonstrated significant inhibition of RABV using an aptamer, (2) tested Lipofectamine RNAi-Max as a means for delivery, and (3) produced significant RABV inhibition at 24 h p.i. This study serves as a proof-of-concept to potentially use aptamers and siRNAs as rabies immunoglobulin (RIG) replacements or therapeutic options for RABV and provides strong evidence towards their further investigation.

## 1. Introduction

A global target for dog-mediated human rabies elimination has been set for the year 2030 [1]. A country-centric approach that applies strategic mass dog vaccination has been identified as the most effective and feasible means to this end, as it focuses on eliminating the disease at its source, thereby reducing the numbers of exposed humans over time [1,2,3]. The prophylactic treatment of exposed individuals provides another means by which the number of human rabies deaths, of which there are tens of thousands every year [4], can be reduced [5]. However, post-exposure prophylaxis (PEP) is not feasible on the mass scale required for the elimination of rabies in humans, as it is a costlier reactive solution as opposed to the proactive solution [6]. Despite this, PEP continues to play a vital role in achieving success through life-saving treatment—in unison with mass dog vaccination—towards achieving the global elimination goal [5].

While rabies PEP comprising both vaccine and rabies immunoglobulin (RIG) is 100% effective if administered correctly, the cost and availability of RIG in particular are a major limitation [7]. The direct and indirect costs associated with an exposed individual receiving adequate PEP can be astronomical in relation to their monthly income, bearing in mind that rabies primarily affects the poorest communities [4]. RIGs are a scarce resource and the world chronically experiences a global shortage of these products [5]. The shortage thus limits the supply of the life-saving RIG to certain health facilities, typically to larger hospitals in urban areas—meaning that those at greatest risk (those in poor and rural communities) are left wanting [7]. Factors that compound the global shortage can be attributed to the high production costs, the requirement for human donors (in the case of human RIG; HRIG), or the specialized production and purification required for equine RIG (ERIG), resulting in the products being exorbitantly costly (ranging between USD25 and 200) and inaccessible to the majority of the rabies-exposed population [7,8,9].

With these challenges and shortcomings being evident, a more cost-effective, mass-producible alternative for RIG has been sought over the past decade. Monoclonal antibodies (mAbs) have shown the most promise, with multiple studies showing their equivalent or superior efficacy compared with RIG [9,10,11]. With these successes, the World Health Organization (WHO) has—for the first time—included their recommendation for the use of certain mAb cocktails as a suitable alternative to RIG [12] and the first mAb has been licensed for use in India [5]. However, despite their success and their relative cost-effectiveness compared to HRIG, there remain many tests and challenges associated with mAb cocktails that will need to be addressed before they become more widely available and accepted as a suitably cost-effective replacement for the current PEP regime [12].

Rabies PEP is only effective before the onset of clinical symptoms and should be initiated without delay [13]. Once clinical symptoms manifest, there is no suitable treatment available and the exposed individual will succumb to the disease in 99% of cases. Many attempts at a feasible treatment have been tested, with the most widely known and tested being that of the “Milwaukee protocol,” first used in 2004 to save the life of an exposed person [14]. However, after more than 70 re-attempts, and several iterations of the protocol (version 6 at the time this manuscript was written), there have been only a handful of survivors from rabies (*n* = 14)—a figure widely refuted in the scientific community [15,16,17]. Currently, the WHO recsommends intensive palliative care for symptomatic individuals and does not recognize any other treatment option as being clearly effective [5]. With these shortcomings, coupled with those currently being experienced with RIG and rabies PEP, research continues into the prospect of identifying a suitable RIG replacement and possible therapeutic for the treatment of RABV.

Rabies virus (RABV) genomes contain five genes, each of which encodes multi-functional proteins with the ability to affect the host immune response in some way to aid the replication and spread of the virus [18]. The importance of these immune-suppressive effects is evident when comparing laboratory-attenuated strains to wild-type RABVs, as the former lack many of these immune-suppressive capabilities, enabling the virus to be cleared from the central nervous system (CNS) by the host’s immune response [19]. Furthermore, studies have demonstrated that it is feasible to clear wild-type RABV infection from the CNS when the host immune response remains unimpeded [20]. Thus, by targeting the immune-suppressive effects of RABV, it may be feasible to create a therapeutic capable of clearing an active infection after the onset of symptoms.

Short-interfering RNAs (siRNAs) are short, double-stranded RNA (dsRNA) molecules that are involved in the naturally occurring RNA interference (RNAi) pathway present in eukaryotic cells [21]. When activated, long dsRNAs are processed by the Dicer enzyme into siRNAs, typically between 21 and 23 nucleotides in length [22,23,24]. siRNAs are incorporated into the RNA-induced silencing complex (RISC), where the antisense strand is used to bind complementary mRNAs, which are subsequently cleaved and degraded [25,26,27]. This process can be repeated multiple times, as the RISC remains stable and functional for several days to several weeks, depending on the cell type [28,29,30]. siRNAs can be chemically synthesized and used to target the degradation of specific mRNAs, making them good potential candidates for therapeutics.

One of the major limiting factors for siRNA therapies is the means of delivery, as this affects their stability in serum and their dosage, facilitates target specificity, and improves incorporation into the RISC [31]. Commonly, naked, modified siRNAs have been used with great success, with multiple naked siRNAs in clinical trials [32]. Other common methods of delivery include lipid nanoparticles and conjugated siRNAs, each with their own advantages and disadvantages (reviewed in [32]). For rabies, only 11 studies have been undertaken to determine the potential use of siRNAs in rabies treatment, with little conclusive evidence indicating the viability of this therapeutic for rabies [33]. A recent study has provided the most promising results, with an siRNA (siRNA 360) producing an 80.6% reduction in RABV phosphoprotein (P) mRNA in in vitro trials when treatment occurred at 2 h post infection (p.i.) [34]. Yet, more evidence is required for siRNAs to be widely considered as an option in the pursuit of a RIG alternative or a potential treatment for rabies.

Similarly, aptamers are relatively novel therapeutic options that have recently been gaining more interest not only for the treatment of diseases but also for diagnostic purposes [35,36]. Aptamers are short single-stranded DNA or RNA oligonucleotides that are typically generated through a selection process (known as SELEX) from a random oligonucleotide library [37,38,39]. Aptamers are analogous to antibodies in their function, as they fold into secondary structures that are capable of target-specific binding [35]. However, aptamers have a multitude of advantages over their antibody analogues, including their low immunogenicity, and their chemical selection and synthesis processes [40,41] and have demonstrated great potential in the treatment of cancers and viral infections [42,43,44,45,46]. Many aptamers are commercially available, and more are currently in various stages of clinical trials [40]. As a result of their stringent target specificity, aptamers have also been used successfully as a means for delivery of other therapeutics [35,36]. For rabies, only four studies have delved into the possibilities of using aptamers as RIG replacement or a potential therapeutic [47,48,49,50]. To date, the most promising results have been demonstrated in vivo when 87.5% of mice survived when treated 24 h prior to challenge, and 50% survived when treated at the same time as challenge [48]. Unfortunately, to date, no study has shown any statistically significant inhibition of RABV (in vitro or in vivo) when treatment occurred after viral challenge. Thus, as is the case with siRNAs, further research needs to be undertaken to determine their potential for rabies treatment.

This study sought to provide evidence towards the use of siRNAs and aptamers as suitable alternatives to RIG, or as a possible treatment for RABV after the onset of symptoms, through in vitro trials. siRNAs targeting the RABV nucleoprotein (N) and P genes were developed due to their roles in RABV replication, their high copy numbers (N gene due to transcriptional bias [51]), and their exemplary role in immune suppression and evasion. Aptamers were developed using cell-selection targeting RABV-infected cells as opposed to specific gene products, enabling the recognition of a diverse range of antigens and cell morphologies that present upon infection by RABV. A variety of delivery methods were tested for the siRNAs, and the aptamers were also tested for their viral inhibitory capabilities. Finally, aptamer–siRNA chimera molecules were created to determine whether the combined actions of these two molecules would be greater than the individual components.

## 2. Materials and Methods

### 2.1. siRNA Design and In Vitro Trials

siRNA molecules were designed using the BLOCK-iT RNAi Designer (Thermo-Fischer Scientific) online designer (https://rnaidesigner.thermofisher.com/rnaiexpress/ accessed on 1 September 2017). Three Stealth RNAi siRNAs (Thermo Fisher Scientific, Johannesburg, South Africa) and one scrambled siRNA for each of the N and P genes of the CVS-11 strain of RABV (accession number: GQ918139.1) were designed. The scrambled siRNAs had randomly generated sequences and acted as controls for all experiments. The Stealth siRNAs were reconstituted in nuclease-free water from a 20 nmol yield to a 20 μM stock solution. All the challenge trials were undertaken using 24-well cell culture plates (Nest Biotechnology Co., Rahway, NJ, USA) to ensure consistency and scalability. All experiments were undertaken in sextuplicate (*n* = 6) for statistical purposes.

Twenty-four-well cell culture plates were seeded with 100 μL of Mouse neuroblastoma (MNA) C1300 clone NA cells and 750 μL of supplemented cell culture media (Dulbecco’s Modified Eagle’s Medium Ham’s F-12 with 15 mM Hepes, L-Glutamine (DMEM-F12; BioWhittaker, Lonza, Belgium) supplemented with 10% gamma-irradiated Fetal Bovine Serum (FBS; origin: South America; Gibco by Life Technologies, Carlsbad, CA, USA) and Penicillin/Streptomycin Amphotericin B antibiotics (10,000 U Pen/mL, 10,000 μg Strep/mL, 25 μg Amphotericin B/mL; Lonza)). The plates were incubated for 24 h at 37 °C and 5% CO_2_ (or until 55–70% confluent) to obtain approximately 1.68 × 10^5^ cells per well. Viral stock concentrations were first determined by real-time qRT-PCR using a standard curve [52], and subsequently diluted via serial dilution to achieve either a 5 or 50 multiplicity of infection (MOI) for each experiment. After incubation, 10 μL of Challenge Virus Standard-11 (CVS-11) was added to each well, except for the negative control well where 10 μL of cell culture medium was added. The plates were incubated with shaking for 2 h at 37 °C and 5% CO_2_. The medium was removed and 1 μL of naked siRNA (20 μM) and 750 μL of supplemented cell culture media were added to each well. Each plate was incubated for a further 24 h at 37 °C and 5% CO_2_. After incubation, each well was sampled by first pipetting rigorously to dislodge and lyse cells, and then transferring 750 μL of the supernatant to a corresponding 2 mL lock cap tube containing 700 μL of TRIzol reagent. RNA was extracted using the Direct-zol RNA MiniPrep Plus kit (Zymo Research, Irvine, CA, USA) and subjected to real-time qRT-PCR analysis to determine viral RNA concentrations, as described previously [52].

The same experimental procedure was followed for all siRNA trials, with the only alteration being the addition of various treatment siRNAs (naked or liposome-conjugated, tested at MOIs of 5 and 50). Lipofectamine RNAi-MAX-conjugated siRNAs were prepared prior to their addition as per the manufacturer’s guidelines (Thermo Fisher Scientific, South Africa) with a final siRNA concentration of 5 pM Lipofectamine-siRNA per well. Positive controls consisted of CVS-11 and Lipofectamine RNAi-MAX, while negative controls consisted of cell culture media and Lipofectamine RNAi-MAX. After the addition of the Lipofectamine–siRNA mix, 750 μL of cell culture media was added to each well, after which the plates were incubated for 24 h at 37 °C and 5% CO_2_, as described before. The means of the viral RNA concentrations were determined and subtracted from the mean of the positive control to determine the mean relative difference for each siRNA.

### 2.2. Development of X-Aptamers and In Vitro Trials

A cohort of 30 X-aptamers (XAs) were selected using the “X-Aptamer Selection Kit—Cells” (AM Biotechnologies, LLC, Houston, TX, USA) as per the manufacturer’s instructions. The selection method included positive-selection steps and a negative-selection step to reduce potential non-specific binding of the XAs to uninfected cells. Unsupplemented DMEM-F12 was used as the selection buffer throughout the selection process.

In a similar manner to the siRNA challenge trials, MNA cells were treated with 10 μL of XA (30 nM) 2 h after challenge. Each XA was tested in sextuplicate at both 5 and 50 MOI. Viral RNA concentration and titer were determined using real-time qRT-PCR with a standard curve [52]. The means of the viral RNA concentrations were determined and then subtracted from the mean of the positive control to determine the mean relative difference for each XA.

### 2.3. Formation of Chimera and In Vitro Trials

To determine the cumulative effect of siRNAs and the XAs, four XAs and four siRNAs (Table 1) were selected and tested together, as a mixture as well as in the form of chimeric molecules. For the mixture, 10 μL of the selected unmodified XA (30 nM) and 10 μL of the selected, unmodified naked siRNA (10 μM) was added to each well at 2 h p.i. Each of the selected siRNAs was tested in conjunction with each of the selected XAs (16 combinations). The cells were subsequently incubated for a further 24 h, sampled, and tested as described previously.

Following the evaluation of the combinations of mixtures of siRNA and XA, chimeric molecules were developed. Initially, chimera were generated by mixing 30 pmol of siRNA with 30 pmol XA and 15 pmol avidin in the ratio of 2:2:1 (ChimeraL). The mixture was incubated at room temperature for 10 min to enable binding before use. Each combination of chimera was tested in sextuplicate. Cells were challenged with 50 MOI CVS-11 and incubated for 2 h. After the incubation, 10 μL of the respective chimera was added to each well. Positive control wells consisted of CVS-11 only, and negative control wells were mock-treated with cell culture media. The plates were incubated for 24 h after treatment and sampled as described previously.

After the analysis of the results obtained from the initial chimera challenge trials (ChimeraL), eight of the most effective aptamer–siRNA chimeras were selected to determine whether a higher concentration of each of the molecules would improve the knockdown effects of the chimeras. Therefore, 30 μL of XA (30 nM) were mixed with 30 μL of siRNA (10 μM) and 30 μL of avidin (10 μM) and incubated for 10 min at room temperature. These chimeras were then challenged at either 2 h (Chimera2) or 24 h (Chimera24) post infection.

### 2.4. Statistical Analyses

Statistical analyses were undertaken for each experimental trial using a one-tailed *t*-test: Paired Two Sample for Means. Comparisons for each of the siRNA were done with reference to the positive control, where the means of the results from the real-time qRT-PCR analyses of the six experimental repeats were considered. Calculations were undertaken using the Data Analysis Tool plugin in MS Excel (Microsoft Office 365 ProPlus; Microsoft Corporation). One-tail *P* scores of less than 0.05 (95% confidence) were considered significant. In the case of significant reductions of the scrambled siRNA, a two-tailed *t*-test: Paired Two Sample for Means was undertaken between the mean of the siRNA and the mean of the scrambled siRNA control. Two-tail *P* scores of less than 0.05 (95% confidence) were considered significant. Lastly, a One-way ANOVA with post-hoc Tukey Honestly Significant Difference (HSD) test was undertaken to determine the statistical significance of results between all the mixture and chimera combination trials from the study. The calculations were undertaken using the online tool available at: http://astatsa.com/OneWay_Anova_with_TukeyHSD/ (Navendu Vasavada 2016; freely available under the creative commons license, accessed on 4 December 2017).

## 3. Results

### 3.1. siRNA Challenge Trials

Stealth siRNAs were designed using the BLOCK-iT RNAi Designer (Life Technologies) online design program. Three siRNAs and one scrambled control for both the N and P genes of RABV CVS-11 were designed, synthesized, and tested in vitro (Table 2).

### 3.2. Naked siRNA Challenge Trials

Each of the siRNAs was initially tested in their native form as naked siRNAs at both 5 and 50 MOI. For the trials conducted using naked siRNA at 5 MOI, 3 siRNAs resulted in a minor reduction in viral RNA concentration, with no statistical significance (data not shown). At 50 MOI, 4 of the 8 siRNAs produced statistically significant mean reductions, including siRNAs N8, N1082, N8C and P330, with siRNA N1082 producing the greatest significant mean reduction (45.54%; *p* < 0.05) (Table 3). However, N8C—a scrambled control siRNA—also produced a statistically significant reduction. To test whether the reduction noted by the RABV-specific siRNAs was significantly different from that of the scrambled control, we proceeded to undertake a two-tail *t*-test. The statistical analyses determined that none of the RABV-specific siRNAs produced significant reductions in viral RNA concentrations compared with the scrambled control, indicating that the treatment with naked siRNAs was statistically ineffective (Table 3).

### 3.3. Lipofectamine-siRNA Challenge Trials

In search of improved RABV knockdown, we sought to test the efficacy of the siRNAs using a different means of delivery. We thus proceeded to conjugate each of the siRNAs with Lipofectamine RNAi-MAX—a transfection reagent commonly used for siRNA delivery—and to test these siRNAs against RABV challenge. At 5 MOI, treatment with any of the Lipofectamine–siRNAs—except for P91C, the scrambled control—resulted in a reduction in mean viral RNA concentration. However, only four siRNAs (siRNAs N8, N53, N1082, and P721) showed statistically significant reductions (*p* < 0.05; each being greater than 50% reduction) compared to the mean of the positive controls (Table 3). The other scrambled control, N8C, showed some reduction compared with the positive control, but this reduction was shown to be statistically insignificant (Table 3). Of the four siRNAs with significant reduction, siRNA N8 was the most effective siRNA with a mean reduction of 67.24% (*p* < 0.005). At 50 MOI, improved results were noted across all Lipofectamine-siRNAs tested. Six of the eight siRNAs tested produced statistically significant mean reductions from the mean of the positive control (Table 3). Of these, N8, N53, and N1082 all resulted in a mean reduction of more than 70%, while P330 was the most effective siRNA of those targeting the P gene. However, P91C (scrambled control siRNA) also showed significant reduction from the positive control. Once again, a two-tail *t*-test was undertaken. The results indicated that the reductions in viral titers associated with siRNAs N8, N53, and N1082 were significantly different compared to the scrambled control (*p* < 0.05), but those of P330 were not (Figure 1).

### 3.4. X-Aptamer Challenge Trials

In total, 30 different XAs targeting RABV-infected MNA cells were selected using the “X-Aptamer selection kit—Cells” (AM Biotechnologies, LLC, USA) that utilizes a cell-selection approach as opposed to the more conventional gene-target approach (Table 1).

The XAs that were developed were subsequently tested in vitro at 2 h p.i. in a similar manner to the siRNA challenge trials. At 5 MOI, four of the XAs showed statistically significant reduction in the mean viral titer compared with the mean of the positive control, with the reduction varying from ~31% to ~67% (Figure 2). At 50 MOI, 11 XAs produced a significant decrease in viral titer compared with the mean of the positive control. Reduction in viral titers ranged from 29% to 61%, with XAs UPRET 2.01a, UPRET 2.03, and UPRET 2.15 all producing more than 50% reduction compared with the mean of the positive control—the most effective of which was UPRET 2.03 (Figure 2). These are the only results for aptamers targeting RABV that depicted a statistically significant reduction in viral RNA when administered after challenge with RABV. At both 5 and 50 MOI, some aptamers resulted in a statistically significant increase in viral RNA concentration and were not tested further as potential RABV therapeutic candidates.

### 3.5. X-Aptamer and siRNA Mixture

To determine whether treatment using the most effective siRNAs and XAs in combination with one another would be improved, we sought to create a mixture of each of the selected XAs with each of the selected siRNAs. Four of the most effective siRNAs and four XAs were selected for further analysis, resulting in 16 combinations being tested. Each of the selected naked siRNAs was added simultaneously with each of the selected XAs to RABV-infected cells at 2 h p.i. Based on previous results, XA UPRET 2.07a was tested at 5 MOI, whereas all the other XAs were tested at 50 MOI in sextuplicate. Our results indicated that 11/16 XA-naked siRNA combinations produced significant reduction (*p* < 0.05) compared to the mean of the positive control. The percentage viral reduction ranged between 34 and 56%, with the combination of UPRET 2.15 + N8 resulting in the greatest reduction (55.82%; *p* < 0.01) compared with the mean of the positive control (Figure 3). The combination of UPRET 2.03 + N53 resulted in a 55.55% reduction in mean viral titer (*p* < 0.005).

### 3.6. X-Aptamer–siRNA Chimera at Lower Concentrations (ChimeraL)

Following from the previous results obtained from mixing the naked siRNAs with the XAs, we proceeded to determine whether the conjugation of the siRNAs and XAs would result in improved knockdown capabilities with the hypothesis that the XAs would serve as target-specific delivery vehicles for the siRNAs. The chimera were formed in a 2:2:1 ratio (XA:siRNA:avidin) as described previously [53]. The newly formed chimeras were named ChimeraL and were tested at 50 MOI only. The ChimeraL with the greatest knockdown effect showed a mean reduction of ~63% (UPRET 2.03–N53; *p* < 0.005) compared with the mean of the positive control, whereas the same combination without the avidin linker resulted in only a 55.55% reduction in mean viral RNA concentration. Contrary to our hypothesis, some XA–siRNA combinations showed higher viral inhibition when simply mixed compared with the chimera (Figure 3).

### 3.7. X-Aptamer–siRNA Chimera at Working Concentrations (Chimera2)

Due to the lack of significant difference noted between the “mixed” trials and the ChimeraL trials, we hypothesized that the therapeutic concentrations were too low to successfully inhibit replication. We thus proceeded to test the eight best chimeras from the ChimeraL trials at higher concentrations, using the concentrations used initially during the testing of the siRNAs or XAs individually. Viral knockdown ranged from 37.8% to 58.3% between the chimeras, with three chimeras (UPRET2.01-N8, UPRET2.03-N8, and UPRET2.15–N53) producing statistically significant results (*p* < 0.05). To compare results, we undertook statistical analyses using a One-way ANOVA with post hoc Tukey HSD test and noted no significant difference between the higher concentration chimeras (Chimera2) and the lower concentration chimeras (ChimeraL) (Table 4).

### 3.8. X-Aptamer–siRNA Chimera at Working Concentrations, with Treatment 24-h p.i.

As a two-hour delay between viral challenge and treatment is not typically representative of the average exposure in a real-life setting (i.e., bite exposure and treatment-seeking behavior), and to test whether the effects of chimera on viral replication were delayed, we proceeded to test the Chimera2 set at 24 h p.i. (Chimera24). At 24 h p.i., two of the chimeras produced statistically significant reductions (36.5% and 23.2%; *p* < 0.05) compared with the mean of the positive control (UPRET 2.01–N8 and UPRET 2.01–N53) (Figure 4). Statistical analyses indicated that the Chimera24 trials were less effective than any of the other combinations (*p* < 0.01), as expected based on the later time point.

In a comparative analysis of each of the combination approaches, we determined that there was no significant difference between the mixed molecules challenged at 50 MOI and any of the chimeras, or any of the chimera with one another, except for the Chimera24 treatment that was significantly reduced (Table 4).

## 4. Discussion

In the last few years, significant strides have been made with regard to rabies control and elimination through improved awareness and disease prioritization; the development and implementation of novel tools [2,54,55]; the formation of standardized, One Health, rabies-specific networks [56,57]; and a global strategic plan developed by the United Against Rabies collaboration [1]. Despite recent improvements to recommendations for human rabies prophylaxis and treatment, there remain many challenges relating to adequate PEP delivery, including disease awareness in communities, dissemination of biologics to these communities, and the high cost of rabies PEP—with special mention of RIG [4,7]. Therefore, effective but cheaper and more readily available alternatives to RIG would have a significant impact on efforts towards equitable rabies prophylaxis across the marginalized dog rabies-endemic regions of the world. In addition, there is still no effective treatment for rabies once the onset of symptoms occurs. Treatment for rabies is complicated by the neurotropism of the virus (sequestered spread and replication in the peripheral and central nervous systems) combined with the evolutionary combative means of the virus to suppress the host’s immune response. As evidenced in laboratory trials, RABV can be cleared from the infected host in various circumstances, including when an adequate immune response is launched (especially in the case of protective antibodies) or when a RABV strain is non-pathogenic (in the case of laboratory-attenuated strains or non-pathogenic strains) [20,58]. With this evidence, it can be conceived that rabies infection (after the onset of symptoms) can be cleared and the infected host saved, so long as the correct circumstances and treatments are available. Therefore, in this study, we designed therapeutics with the aim of targeting not only RABV replication, but also the immune-suppressive effects of RABV on the host, with the hypothesis that the infected host would be able to launch a suitable immune response to clear the infection.

As siRNAs are efficient, stable, and highly target-specific, they provide the ideal candidate to actively and persistently inhibit RABV replication and spread. Unfortunately, the few studies undertaken for rabies have thus far demonstrated limited success [33]. The most recent study by Ono and colleagues (2017) demonstrated the capability of achieving an 80% reduction in viral titer in vitro with the siRNAs transfected using Lipofectamine 2000 but no statistically significant reduction was shown in vivo [34]. This demonstrated the feasibility of RABV inhibition using siRNAs, but highlighted the current shortcomings of effective, targeted delivery for successful in vivo application. In our study, two different siRNA-specific delivery methods were initially tested and compared—naked siRNAs and Lipofectamine RNAi-Max-conjugated siRNAs. This was the first study to test Lipofectamine RNAi-Max as a delivery mechanism for RABV siRNAs. The naked siRNAs showed no statistically significant inhibition, once again highlighting the importance of effective cytoplasmic entry of the siRNA into the target cell. In contrast, the Lipofectamine RNAi-Max conjugated siRNAs showed significant knockdown at both 5 and 50 MOI. Three siRNAs (siRNAs N8, N53, N1082) showed statistically significant reductions (*p* < 0.05; each being greater than a 50% reduction), with siRNA N8 being the most effective siRNA (mean reduction of 67.24%; *p* < 0.005). At a higher MOI, results were further improved, with knockdown levels reaching as high as 80.13% for siRNA N53, while N8 and N1082 also resulted in more than a 70% reduction in viral RNA. The inhibition of viral RNA with siRNA N53 is one of the most effective knockdowns of RABV using siRNAs demonstrated to date—equivalent to the results obtained in a prior study [34]. Furthermore, Lipofectamine RNAi-Max greatly improved efficacy, likely through more efficient delivery, demonstrating the crucial role of effective delivery in siRNA efficacy.

Aptamers have also shown promise as therapeutic agents, with many comparisons to mABs being evident. However, only limited studies have been undertaken to determine their feasibility as therapeutic agents against RABV. For other diseases, aptamers have been utilized for targeted delivery of therapeutic agents due to their target-specific nature and ability to locate and bind specific epitopes [59]. This, coupled with their scalability and cost-effectiveness for production, makes them an ideal candidate to facilitate the targeted delivery of site-specific siRNAs. Previous studies for the use of aptamers against RABV have demonstrated limited efficacy, with inhibition only occurring with pre-treatment or treatment at the time of infection. Results such as these are insufficient to warrant further interest or investment into aptamers as potential therapeutics for rabies, as pre-treatment is infeasible for the elimination of dog-mediated human rabies. However, our study was the first to show significant reductions in viral RNA concentrations with aptamer treatment p.i. XAs UPRET 2.01a (58.43%), UPRET 2.03 (61.3%) and UPRET 215 (53.56%) are the most effective aptamers against RABV in a p.i. setting to date. Crucially, the XAs were designed to target RABV-infected cells rather than specific gene targets, making them potentially more diverse and applicable to various RABV strains and other RABV variants. Furthermore, their ability to directly target infected cells created an opportunity to use these molecules not only in targeted therapeutics, but also for targeted delivery and potentially for diagnostic purposes. Thus, despite the focus in this study being primarily on their therapeutic potential, we believe that the opportunities to investigate aptamers further would be fruitful for progressive diagnostic and therapeutic techniques in the field of rabies. Lastly, the novel methodology developed in this study in conjunction with AM Biotechnologies LLC for the rapid and cost-effective selection of suitable XAs provides further opportunities for continued research into aptamers for RABV as well as other potential targets. Previously, aptamer selection was a lengthy, challenging, and costly approach, but with the development of this selection method, the time taken and cost associated with the selection of effective aptamers have been significantly reduced.

Despite the siRNAs being highly effective in inhibition and reduction of RABV viral RNA, further improvement was required for these therapeutics to be competitive with mAbs for RABV treatment and as a RIG replacement. Evidence from this study and those prior suggested that the means of effective delivery remained the limiting factor. Therefore, because there have been few investigations into the use of aptamers or siRNAs against RABV, because these agents have never been tested concomitantly against RABV, and because of the express need for a suitable RIG alternative and treatment for rabies, we aimed to test siRNAs and XAs concomitantly to provide support for the use of these molecules in RABV prophylaxis and treatment. Thus, we sought to improve targeted siRNA delivery directly to infected cells using the highly specific XAs.

To test whether there were any cumulative effects or negative interactions between the siRNAs and XAs, we tested these molecules in a mixture. The most effective mixture was that of UPRET 2.15 + N8 resulting in a 55.82% reduction. In comparison with the XA alone (53.56%) and naked siRNA alone (35.2%), an overall increased effect was noted. However, in comparison with the same Lipofectamine-siRNA (72.4%), the results were diminished. This suggested that the use of Lipofectamine was superior to that of the XA + siRNA mixture tested—an anticipated finding as the XAs in the mixture would not facilitate targeted siRNA delivery as they were not conjugated. We then proceeded to develop XA–siRNA chimera, but these resulted in no significant difference to the same siRNAs and XAs tested in a mixture at both a lower and a higher working concentration. Therefore, the conjugation of the XAs and siRNAs was seemingly ineffective in improving siRNA delivery. This could be attributed to the means of conjugation used, as studies for other therapeutic targets have demonstrated success utilizing a variety of conjugation/linkage methods [60]. Furthermore, it must be taken into consideration that the XA was bound to the surface of the infected cell and was required to release the siRNA for uptake into the cytoplasm. Unfortunately, this study did not explore the mechanism of action for the binding of the XA and whether the siRNA was in fact released into the cytoplasm. Therefore, to further improve siRNA efficacy, we believe that alternative binding methods and the mechanism of action pertaining to the XA binding and siRNA release should be elucidated, ensuring that the siRNA is in fact delivered to the target cell and that it is subsequently released for efficient uptake into the cytoplasm.

Despite the limited success of the chimera at 2 h p.i., this study demonstrated the most effective, statistically significant inhibition of RABV using siRNAs or aptamers at 24 h p.i. to date. With the results described in this study, we believe that both siRNAs and aptamers have the potential to be feasible, cost-effective candidates as a RIG replacement. Furthermore, targeting specific aspects of immune evasion and suppression mechanisms of the RABV, coupled with the targeted degradation of viral RNA, is potentially a feasible approach to developing a treatment for RABV-infected individuals after the onset of symptoms. Considering that siRNAs have a long-term effect [28,29,30], this alone could inhibit viral replication and the immune-suppressive effects of RABV in the host for a sufficient period to enable an adequate immune response to clear the viral insult. Coupling this efficacy with improved, targeted delivery—using aptamers or another highly target-specific molecule such as antibodies or nanoparticles—could provide the solution not only as a RIG alternative through direct injection into the wound site, but also as a potential therapeutic after the onset of symptoms. Using this approach, one could target immune-privileged areas to inhibit replication and further spread and enable subsequent clearance from the CNS by the immune response. Although the use of aptamers in this study as delivery vehicles for the siRNAs was less effective than Lipofectamine RNAi-Max, the proof-of-concept remains, considering the success demonstrated in studies for other diseases [60]. This study had several limitations that would need to be addressed before any conclusions could be made in terms of the potential for siRNAs and aptamers to be used in rabies treatment. The non-specific effects of the scrambled siRNA would warrant further investigation, as based on the statistical results, the addition of certain short RNA sequences elucidated some viral inhibition, despite not being target specific. This viral inhibition could be due to the stimulation of an innate immune response through the binding of the short RNA with toll-like receptors [61]. This includes elucidating the mechanism of action and binding epitopes of the XAs developed, so that their specificity and effectiveness can be better understood. Although this study was a proof-of-principle study, further investigation into the ability of these siRNAs and XAs to target other strains and variants of RABV would be required if these are to be applied as a rabies therapeutic globally. Future studies should investigate the effects of these molecules using additional methods such as viral titration and eventually in vivo models (if applicable). The potential therapeutics developed in this study could not only be further refined beyond proof-of-principle but should subsequently be tested in different cells and animal models to better understand the interactions with immune responses and the overall efficacy of the molecules. However, as a proof-of-principle, this study provides a solid foundation for further research in this field of interest.

A focus on mass dog vaccination is required to eliminate human deaths from dog rabies. However, while working towards this goal, millions of bite cases and rabies exposures will continue to occur. Protective treatment for these exposed individuals will continue to be vital and the search for improvements in the biologics, availability, and affordability for rabies treatment remains a worthwhile objective.

## Figures and Tables

**Figure 1 viruses-13-00881-f001:**
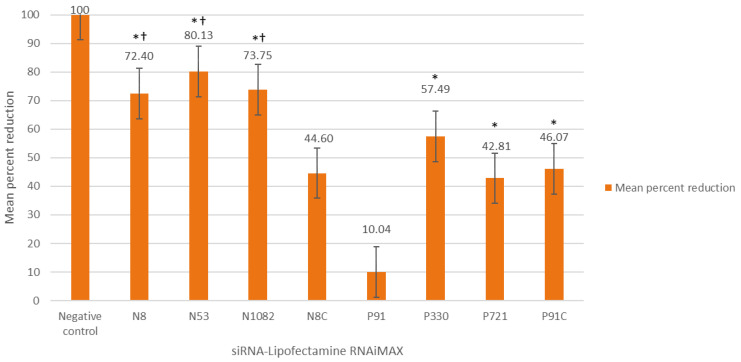
Mean percent reduction (*Y*-axis) of siRNAs conjugated with Lipofectamine (final concentration 5 pM) targeting the N and P genes of RABV CVS-11 (50 MOI) with treatment occurring 2 h post infection. All of the experiments were undertaken in sextuplicate. * indicates significant reduction compared to the mean of the positive control (CVS-11 and Lipofectamine RNAi-MAX). N8 control (N8C) and P91 control (P91C) were scrambled siRNA sequences. † indicates significant difference (*p* < 0.05) of the siRNAs compared with the scrambled control (P91C). Only P91C was compared in the two-tail *t*-test as this scrambled control demonstrated non-specific reduction compared with the positive control. For display purposes, the negative control has been set in relation to the positive control, but no reduction was observed for the negative control.

**Figure 2 viruses-13-00881-f002:**
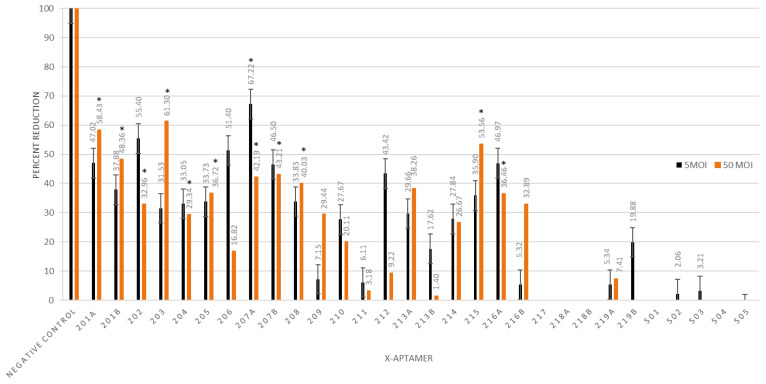
Mean percent reduction (*Y*-axis) of X-aptamers (30 nM) targeting RABV CVS-11-infected cells (5 and 50 MOI) with treatment occurring 2 h post infection. All of the experiments were undertaken in sextuplicate. * indicates significant reduction (*p* < 0.05) compared to the mean of the positive control. For display purposes, the negative control has been set in relation to the positive control, but no reduction was observed for the negative control. Chimera challenge trials.

**Figure 3 viruses-13-00881-f003:**
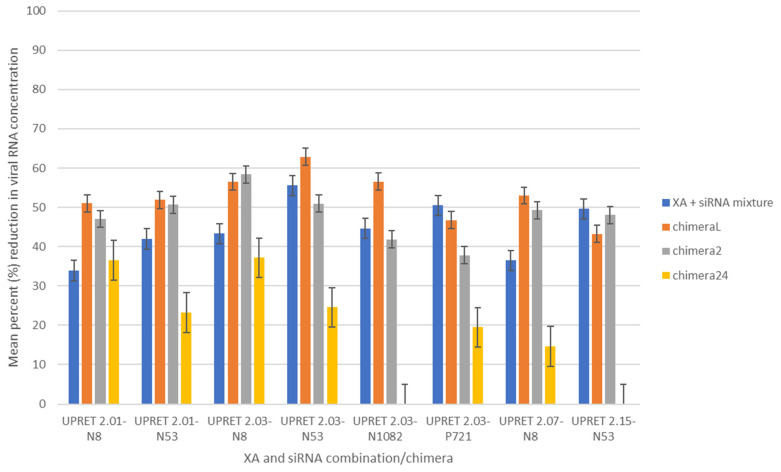
Mean percent reduction (Y-axis) comparison of the XA + siRNA mixture and each of the variations of the chimera (X-aptamer–avidin–siRNA) experiments (ChimeraL, low concentration; Chimera2, working concentration; and Chimera24, working concentration with treatment 24 h after challenge) targeting RABV CVS-11-infected cells. The working concentrations for Chimera2 and Chimera24 were 30 nM XA and 10 μM siRNA. All treatments were tested at 2 h post infection, except for Chimera24 that was tested at 24 h post infection. All the experiments were undertaken in sextuplicate. The mean of the results from the Chimera24 trials was statistically significantly different compared to those of all the other trials (*p* < 0.01). There was no significant difference between any of the other trials.

**Figure 4 viruses-13-00881-f004:**
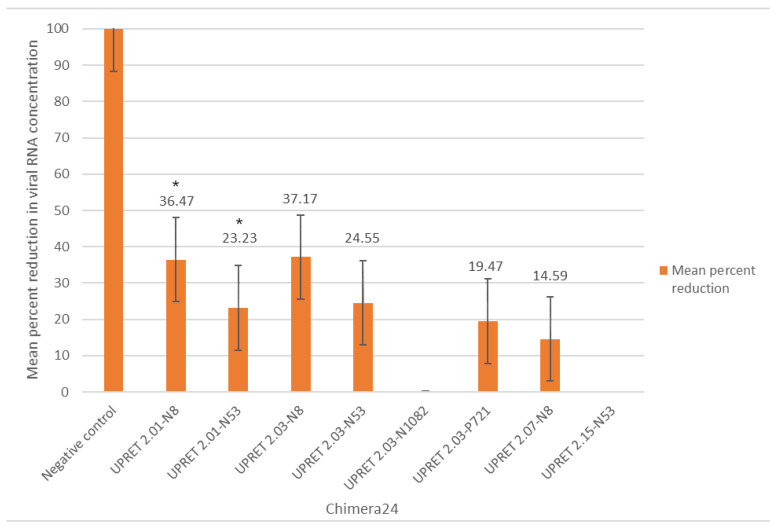
Mean percent reduction (*Y*-axis) of chimera (X-aptamer–avidin–siRNA) targeting RABV CVS-11-infected cells with treatment occurring 24 h p.i. These chimeras (Chimera24) were tested with the same concentration as Chimera2 (30 nM XA and 10 M siRNA). All the experiments were undertaken in sextuplicate. * indicates significant reduction (*p* < 0.05) compared to the mean of the positive control.

**Table 1 viruses-13-00881-t001:** Selected X-aptamers and siRNAs used as mixture combinations as well as for chimerization through biotin–avidin linkage followed by in vitro p.i. trials.

X-Aptamer	siRNA
UPRET 2.01a	N8
UPRET 2.03	N53
UPRET 2.07a	N1082
UPRET 2.15	P721

**Table 2 viruses-13-00881-t002:** Developed Stealth siRNAs targeting the N and P genes of RABV CVS-11.

Name	Sense (5′–3′)	Antisense (5′–3′)	Binding Position on CVS-11 (Accession Number: GQ918139.1)
N8	CCG ACA AGA UUG UGU UCA AAG UCA A	UUG ACU UUG AAC ACA AUC UUG UCG G	78–102
N53	AGC CUG AGA UUA UCG UGG AUC AAU A	UAU UGA UCC ACG AUA AUC UCA GGC U	123–147
N1082	GAG ACG AGA AAG AAC UUC AAG AAU A	UAU UCU UGA AGU UCU UUC UCG UCU C	1152–1176
N8Control (N8C)	UUG GUC AUU UAA GCA CUA AUU CCG G	CCG GAA UUA GUG CUU AAA UGA CCA A	N/A
P91	CCG AAG AGA CUG UUG AUC UGA UCA A	UUG AUC AGA UCA ACA GUC UCU UCG G	1575–1599
P330	GGA GUC CAA AUA GUC AGA CAA AUG A	UCA UUU GUC UGA CUA UUU GGA CUC C	1814–1838
P721	AGA UGA ACC UUG AUG ACA UAG UUA A	UUA ACU AUG UCA UCA AGG UUC AUC U	2205–2229
P91Control (P91C)	CCG AGA GUG UCU AGU AGU CUA ACA A	UUG UUA GAC UAC UAG ACA CUC UCG G	N/A

**Table 3 viruses-13-00881-t003:** Summary of the statistically significant mean reductions in viral RNA concentrations of RABV CVS-11 for each siRNA (both naked and using Lipofectamine for transfection) at 5 and 50 MOI.

	siRNA Naked 5 MOI	siRNA Naked 50 MOI	siRNA Lipofectamine 5 MOI	siRNA Lipofectamine 50 MOI
N8	8.14	35.20 *	67.24	72.40
N53	−9.98	23.61	64.22	80.13
N1082	12.38	41.54 *	61.22	73.75
N8C	−6.59	24.49	15.45	44.60
P91	−14.78	31.61	24.12	10.04
P330	17.50	35.18 *	45.19	57.49 *
P721	−24.32	39.92	56.01	42.81 *
P91C	−25.37	32.14	−5.25	46.07

Shaded cells indicate statistically significant (*p* < 0.05) mean reductions compared with the mean of the positive control. Numbers within each cell indicate the mean percent reduction for each siRNA. * indicates siRNA that produced statistically significant reductions (*p* < 0.05) in comparison with the mean of the positive control, yet were not significantly different from the scrambled control; MOI: Multiplicity of Infection.

**Table 4 viruses-13-00881-t004:** Statistical comparisons of treatment options tested in this study to determine the most effective treatment option for RABV in a p.i. scenario. All negative results were removed from the analyses. Shaded cells indicate statistical significance. vs.—versus. ** indicates statistically significant differences between treatment types.

Treatment Pairs	Tukey HSD Q Statistic	Tukey HSD *p*-Value	Tukey HSD Inference
siRNA + Aptamer vs. ChimeraL	3.2791	0.1198351	insignificant
siRNA + Aptamer vs. Chimera2	1.3958	0.7365419	insignificant
siRNA + Aptamer vs. Chimera24	6.8448	0.0010053	** *p* < 0.01
ChimeraL vs. Chimera2	1.8833	0.5471207	insignificant
ChimeraL vs. Chimera24	9.8806	0.0010053	** *p* < 0.01
Chimera2 vs. Chimera24	8.1370	0.0010053	** *p* < 0.01

## Data Availability

The data presented in this study are openly available in UPSpace repository at https://repository.up.ac.za/ under UP postgraduate Scott, Terence Peter or available on request from the TPS.

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
