# Peer review of "Rabies Prophylactic and Treatment Options: An In Vitro Study of siRNA- and Aptamer-Based Therapeutics"

_viruses, 2021, doi:10.3390/v13050881_

Round 1
Reviewer 1 Report
The authors describe their experiences with respect to in-vitro efficacy of their anti-rabies aptamer, siRNA and aptamer-siRNA chimera.
New and interesting is the combination between siRNA and aptamer technology for the rabies topic and also the very practical approach of application 24hrs after pi.
A few questions came up:
Abstract:
Line 12: rabies biologics – from a regulatory perspective are those oligos no "biologics" – so replace with “rabies therapeutics” or something similar.
Line 15-17: chimera 24hrs pi – what is the pi time for N53 and UPRET 2.03 – since the inhibition percentages are competitive? – same conditions? – specify please,
Line 21: RIG – please spell out at the first use and in Abstract
Introduction:
Line 130: “aptamers…..have been used successfully in the treatment of cancers and viral infections [43-47]”
According to my knowledge this is mostly in-vitro data – I would suggest: “have been shown great potential for treatment of cancer and viral infections
Results:
Line 311: “knockdown” since it is Aptamers only (XA) – it should be called “reduction”
Fig. 1: what is the positive control for Fig. 1?
(i) I read in the footnote for Table 1: “Positive control wells consisted of CVS-11 only, and negative control wells were mock-treated with cell culture media.” Does this apply to all experiments? Also Figure 1? Then this should be indicated in the figure legend.
Was the negative control also set in relation to the positive control? Why does the negative control show 100% reduction?
(ii) “N8 control (N8C) and P91 control (P91C) were scrambled siRNA sequences. † indicates significant difference (p<0,05) of the siRNAs when compared with the scrambled control (P91C).”
Why did you compare all results with the P91C control only? Shouldn’t at least the N8 siRNA sequence be compared to the N8C control? Wouldn’t this make more sense?
What are the concentration of the si-RNA – 10 µM?? Please add.
X-axis has not been labeled,
The grey is not good to see (bad contrast).
Fig. 2: (i) Was the negative control also set in relation to the positive control? Why does the negative control show 100% reduction?
(ii) The text reads: “At 5 MOI, 4 of the XAs showed statistically significant reduction in the mean viral titre when compared with the mean of the positive control, with the reduction varying from ~31% - ~67% (figure 2).”I see only one marked as statistically significant: 67%.
(iii) I read in the text: “with XAs UPRET 2.01a, UPRET 2.03 and UPRET 2.15 all producing more than 50% reduction when compared with the mean of the positive control – the most effective of which was UPRET 2.03 (figure 2).” It is figure 4.
What is the final concentration of the applied XAs? 30 nM? Please add.
Please improve the figure legend. Also: the “*” is not clear to see.
Fig. 3 What is "low concentration"? What is the "working concentration"? Has this been defined before? Please clarify and add.
Fig. 4 What is the “initial working concentration of each individual molecule” Please define.
Author Response
Reviewer 1:
The authors describe their experiences with respect to in-vitro efficacy of their anti-rabies aptamer, siRNA and aptamer-siRNA chimera.
New and interesting is the combination between siRNA and aptamer technology for the rabies topic and also the very practical approach of application 24hrs after pi.
Response to reviewer’s comments:
The authors would like to thank the reviewer for their comments and suggestions towards the improvement of the manuscript. We value your input and have addressed your concerns and suggestions point-by-point below.
Abstract:
Line 12: rabies biologics – from a regulatory perspective are those oligos no "biologics" – so replace with “rabies therapeutics” or something similar.
The word “biologics” has been changed to “therapeutics” as suggested.
Line 15-17: chimera 24hrs pi – what is the pi time for N53 and UPRET 2.03 – since the inhibition percentages are competitive? – same conditions? – specify please,
Thank you for pointing this out. The text was unclear and has been amended according to your suggestion. These therapeutics were tested at 2 hours p.i. in accordance with all other therapeutics unless otherwise specified. The text in the abstract now includes “at 2 hours p.i.” to clarify this.
Line 21: RIG – please spell out at the first use and in Abstract
Thank you. This has been amended accordingly.
Introduction:
Line 130: “aptamers…have been used successfully in the treatment of cancers and viral infections [43-47]”
According to my knowledge this is mostly in-vitro data – I would suggest: “have been shown great potential for treatment of cancer and viral infections.
Thank you. We have amended the sentence as follows: “have demonstrated great potential in the treatment of cancers and viral infections [43–47].”
Results:
Line 311: “knockdown” since it is Aptamers only (XA) – it should be called “reduction”.
Thank you. This has been amended accordingly.
Fig. 1:
What is the positive control for Fig. 1?
Positive controls consisted of CVS-11 and lipofectamine RNAiMAX, whilst negative controls consisted of cell culture media and Lipofectamine RNAiMAX. This has been included into the legend as follows:
* indicates significant reduction compared to the mean of the positive control (CVS-11 and lipofectamine RNAiMAX).
(i) I read in the footnote for Table 1: “Positive control wells consisted of CVS-11 only, and negative control wells were mock-treated with cell culture media.” Does this apply to all experiments? Also Figure 1? Then this should be indicated in the figure legend.
No, this does not apply to all experiments. This applies to all experiments except those that included the lipofectamine, as for these experiments, positive controls consisted of CVS-11 and lipofectamine RNAiMAX, whilst negative controls consisted of cell culture media and Lipofectamine RNAiMAX. Therefore, the positive control for Figure 1 is CVS-11 and lipofectamine RNAiMAX. This has been clarified in the figure legend as mentioned above.
Was the negative control also set in relation to the positive control? Why does the negative control show 100% reduction?
Yes, the negative control was set in relation to the positive control in terms of displaying it on the graph. As we believe it was important to demonstrate the negative control on the graph, the only feasible means would be to set it in relation to the positive control (as showing 100% reduction). This has been clarified in the figure legend with the following sentence: “For display purposes, the negative control has been set in relation to the positive control, but no reduction was observed for the negative control.”
(ii) “N8 control (N8C) and P91 control (P91C) were scrambled siRNA sequences. † indicates significant difference (p<0,05) of the siRNAs when compared with the scrambled control (P91C).”
Why did you compare all results with the P91C control only? Shouldn’t at least the N8 siRNA sequence be compared to the N8C control? Wouldn’t this make more sense?
The results were compared with both N8C and P91C, but N8C showed no statistically significant reduction when compared with the positive control, while P91C showed statistically significant reduction in comparison to the mean of the positive controls. Therefore, a two-tail T-test was undertaken to determine whether the reduction from the RABV-specific siRNAs was significantly greater than that of the scrambled RNA sequence that demonstrated some reduction when compared with the positive control (P91C). Table 3 describes this in more detail. The text in the figure legend has been amended as follows to clarify this: “Only P91C was compared in the two-tail T-test as this scrambled control demonstrated non-specific reduction when compared with the positive control.”
What are the concentration of the si-RNA – 10 µM?? Please add.
The concentration of siRNA before conjugation with the Lipofectamine RNAiMAX was 10µM, but the final siRNA concentration was 5 pM of conjugated lipofectamine-siRNA per well. This has been included into the figure legend for clarification as follows: “Mean percent reduction of siRNAs conjugated with lipofectamine (final concentration 5pM)”
X-axis has not been labelled.
Thank you for identifying this. The whole figure has been changed and improved as suggested, including the labelling of the x-axis.
The grey is not good to see (bad contrast).
Thank you for identifying this. The whole figure has been changed and improved as suggested, including the colour contrast.
Fig. 2:
(i) Was the negative control also set in relation to the positive control? Why does the negative control show 100% reduction?
Yes, this is correct. The following sentence has been included into the figure legend to clarify: “For display purposes, the negative control has been set in relation to the positive control, but no reduction was observed for the negative control.”
(ii) The text reads: “At 5 MOI, 4 of the XAs showed statistically significant reduction in the mean viral titre when compared with the mean of the positive control, with the reduction varying from ~31% - ~67% (figure 2).”I see only one marked as statistically significant: 67%.
Apologies. It seems that the asterisks had shifted and therefore were not clear. The figure has been amended overall to improve clarity and contrast as suggested.
(iii) I read in the text: “with XAs UPRET 2.01a, UPRET 2.03 and UPRET 2.15 all producing more than 50% reduction when compared with the mean of the positive control – the most effective of which was UPRET 2.03 (figure 2).” It is figure 4
Thank you for the suggestion. Figure 2 relates to the reduction observed for the X-Aptamers at 2 hours p.i. The sentence quoted above refers to the reduction for the XAs at 2 hours p.i., whilst Figure 4 refers to the reduction for the chimera (XA+siRNA) at 24 hours p.i.
What is the final concentration of the applied XAs? 30 nM? Please add.
Thank you. This has been added into the figure legend.
Please improve the figure legend. Also: the “*” is not clear to see.
Apologies. It seems that the asterisks had shifted and therefore were not clear. The figure has been amended overall to improve clarity and contrast as suggested. The figure legend has also been improved based on the reviewer’s suggestions.
Fig. 3
What is "low concentration"? What is the "working concentration"? Has this been defined before? Please clarify and add.
Thank you for this. Yes, this had been defined previously in the methods, but clarification has been included into the figure legend as follows: “The working concentrations for Chimera2 and Chimera24 were 30nM XA and 10 μM siRNA.”
Fig. 4
What is the “initial working concentration of each individual molecule” Please define.
Thank you for the feedback. This in fact was a mistake and has been rectified with the following text: “These chimeras (Chimera24) were tested with the same concentration as Chimera2 (30nM XA and 10M siRNA).”

Reviewer 2 Report
General comment
In their study entitled “Rabies prophylactic and treatment options: an in vitro study of siRNA- and aptamer-based therapeutics”, the authors describe the development of short-interfering RNAs (siRNAs) and aptamers against rabies virus (CVS strain), and the in vitro validation of them, alone or in combination. As underlined by the authors, the application of siRNAs and aptamers as antiviral components has been poorly investigation for rabies so far, while they present numerous advantages and promising results in some other diseases. In particular, the authors identified one siRNA (N53) and a combination of aptamer-siRNA (chimera UPRET 2.03-N8) with inhibitory action on rabies virus replication, tested 24h post-infection.
However, I have major comments regarding the methodology and the presentation of the results. In addition, the discussion could be improved regarding the limitation of the study, the next steps based on these results and how these compounds could be used in human in the rabies context. Lastly, even if the manuscript remains clear and well-written as a whole, it remains very long, with some redundancies, and thus would benefit to be reduce. Details are described below.
Major comments
As indicated previously, I find the manuscript well written but it remains too long, and need to be simplified/shortened. As examples:
- Introduction section: the following paragraphs can be reduced: lines 65-76 (on Milwaukee protocol), lines 90-106 (on siRNA) and lines 142-152 (which needs to be simplified by few sentences describing clearly the objectives of this study).
- Materials and Methods section: simplified the “statistical analyses” section (delete the redundancies and/or obvious elements, such as “A null hypothesis of zero mean difference between treatment and positive control was used” lines 235-236, 241).
- Discussion section: this section is very long and need to be reduced (in order to prevent the reader from being lost), by avoiding repetition (e.g. lines 389-417, already largely described in the Introduction section), briefly summarizing the results (e.g. 482-492).
I have some concerns about the methodology used. In particular:
- The authors used negative siRNA controls (scrambled siRNAs), which exhibited sometimes some “non specific” antiviral effects (Table 1). However, they did not use any negative controls with aptamers, which does not make it possible to ensure that the effects observed with these products are specific.
- I am not comfortable with the association between MOI and molecular quantification. Indeed, MOI is based on the measurement of infectious particles, whereas viral quantification based on RTqPCR looks at the presence of RNA (genomic, mRNA, from defective particles, etc.). I suggest to modify it.
- The lack of viral titration (by FFU/mL for example), done in parallel with RTqPCR, is a weakness in this study. These results will be quite interesting to validate the effects of siRNA and aptamers.
- Understanding of siRNA results can be sometimes a bit challenging, regarding the interpretation with those obtained with scrambled controls. Why not trying to normalize directly the results obtained with the specific siRNAs with these controls (after comparison with the positive control), and/or simplified the presentation of these results (e.g. with Table 3 and Figure 1)?
I suggest to modify, throughout the document, the use of “post-exposure treatment” (e.g. line 214) or “not typically representative of the average exposure” (lines 360-361), which I do not consider suitable for in vitro studies. Simplify if with “post infection” for example.
In the discussion section, I suggest also to the authors to discuss more about the potential limitation of this approach in terms of spectrum of antiviral activity. Indeed, siRNAs and aptamers were designed against the CVS strain. What will be the effect on other and most divergent RABV, even other lyssaviruses? Do the authors select conserved regions in the N and P proteins?
Similarly, I did not see any discussion about the viral inhibition effects of the negative siRNA controls. Is it common in such approach? What about the absence of negative controls with the aptamers and the mixed assays?
In addition, minor comments are indicated below:
Minor comments
- Line 11: Develop “siRNA”.
- Line 21: Develop “RIG”.
- Line 63: If I am not mistaken, this timeframe is not anymore indicated in the 3rd WHO report, replace by “People with WHO category II or III exposures should receive PEP without delay as an emergency procedure”. The same with the WER 2018 (No 16, 2018, 93, 201–220) with “promptly started after an exposure”. I suggest to modify accordingly.
- Lines 164-165: How many cells were used per well?
- Line 216 (and for all the manuscript): Replace “pMol” with “pmol” (quantity) or “pM” (concentration).
- Figure 1, 2, 3 and 4: Indicate in the legend indicate what the intervals refer to.
- Figure 3: Asterisks and numbers are superimposed, which make the reading difficult.
Author Response
Reviewer 2:
General comment
In their study entitled “Rabies prophylactic and treatment options: an in vitro study of siRNA- and aptamer-based therapeutics”, the authors describe the development of short-interfering RNAs (siRNAs) and aptamers against rabies virus (CVS strain), and the in vitro validation of them, alone or in combination. As underlined by the authors, the application of siRNAs and aptamers as antiviral components has been poorly investigation for rabies so far, while they present numerous advantages and promising results in some other diseases. In particular, the authors identified one siRNA (N53) and a combination of aptamer-siRNA (chimera UPRET 2.03-N8) with inhibitory action on rabies virus replication, tested 24h post-infection.
However, I have major comments regarding the methodology and the presentation of the results. In addition, the discussion could be improved regarding the limitation of the study, the next steps based on these results and how these compounds could be used in human in the rabies context. Lastly, even if the manuscript remains clear and well-written as a whole, it remains very long, with some redundancies, and thus would benefit to be reduce. Details are described below.
Major comments
As indicated previously, I find the manuscript well written, but it remains too long, and need to be simplified/shortened. As examples:
- Introduction section: the following paragraphs can be reduced: lines 65-76 (on Milwaukee protocol), lines 90-106 (on siRNA) and lines 142-152 (which needs to be simplified by few sentences describing clearly the objectives of this study).
- Materials and Methods section: simplified the “statistical analyses” section (delete the redundancies and/or obvious elements, such as “A null hypothesis of zero mean difference between treatment and positive control was used” lines 235-236, 241).
- Discussion section: this section is very long and need to be reduced (in order to prevent the reader from being lost), by avoiding repetition (e.g. lines 389-417, already largely described in the Introduction section), briefly summarizing the results (e.g. 482-492).
Response to reviewer 2:
The authors would like to thank the reviewer for their valuable feedback and comprehensive review. We appreciate the time taken to provide the feedback that was given to improve the manuscript and clarify the outcomes.
In terms of the major comments provided, we acknowledge that the manuscript was lengthy and sometimes repetitive and have worked to significantly shorten both the entire introduction as well as the discussion sections based on the reviewer’s feedback and our own revision with the reviewer’s comments in mind. In addition, we have simplified the statistical analyses section as per the reviewer’s recommendations. We hope that with these changes, the manuscript is clearer and easier to follow.
In terms of the questions regarding the methodology and the minor comments, we have addressed those point-by-point below.
I have some concerns about the methodology used. In particular:
- The authors used negative siRNA controls (scrambled siRNAs), which exhibited sometimes some “non specific” antiviral effects (Table 1). However, they did not use any negative controls with aptamers, which does not make it possible to ensure that the effects observed with these products are specific.
Yes, it is correct that we did not use any random aptamer controls (the negative control was media only), but this was due to the nature of the aptamers developed. The aptamers were developed using a method equivalent to “cell-SELEX” meaning that the aptamers were selected using a random DNA library that bound selectively to the RABV-infected cell. As any random DNA could potentially have inhibitory effects (based on the targeting of unknown or novel epitopes on infected cells, etc.) it would not have been possible to generate a truly negative control without determining the exact binding mechanism and target epitope of every random DNA molecule. Even by including unbound DNA, this may have had a different and unexpected resultant effect on the experiments, such as stimulating immune responses or binding to the selected X-aptamers themselves.
To address this, the authors have included an additional sentence into the discussion section that reads as follows: “This includes elucidating the mechanism of action and binding epitopes of the XAs developed, so that their specificity and effectiveness can be better understood.”
- I am not comfortable with the association between MOI and molecular quantification. Indeed, MOI is based on the measurement of infectious particles, whereas viral quantification based on RTqPCR looks at the presence of RNA (genomic, mRNA, from defective particles, etc.). I suggest to modify it.
The authors agree that qRT-PCR is typically not a good correlate for multiplicity of infection, but in this case, an external standard curve was used to make this correlation. For the standard curve, the viral RNA concentration was first determined based on the quantitative real-time PCR results. From these results depicting the RNA concentrations, an external standard curve was developed as described previously in an article by Coertse et al 2010. This external standard curve was developed as a specific method to quantify viral titres. The standard curve was developed by determining the TCID50 for various viral concentrations of CVS-11 (the same virus isolate as used in this study) in MNA cells (the same cell line as used in this study) under controlled conditions. The TCID50 results were then correlated to the CT values, using the Spearmann-Kaber method, to generate a standard curve. This standard curve was reconfirmed for this study and was subsequently used as a means to determine the viral titre from the Real time qRT-PCR used in this study under the same conditions as the controlled conditions used to generate the standard curve. This is reflected in the text from the methods section as follows: “Viral stock concentrations were first determined by real-time qRT-PCR using a standard curve [53], and subsequently diluted via serial dilution to achieve either a 5 or 50 MOI for each experiment.” For this reason, we have chosen to leave the text unaltered.
- The lack of viral titration (by FFU/mL for example), done in parallel with RTqPCR, is a weakness in this study. These results will be quite interesting to validate the effects of siRNA and aptamers.
The authors fully agree with this statement from the reviewer. We acknowledge this as a clear limitation of the study. This has been highlighted in a new addition to the discussion highlighting shortcomings, limitations, and future recommendations. For example: “Future studies should investigate the effects of these molecules using additional methods such as viral titration and eventually in vivo models (if applicable).”
- Understanding of siRNA results can be sometimes a bit challenging, regarding the interpretation with those obtained with scrambled controls. Why not trying to normalize directly the results obtained with the specific siRNAs with these controls (after comparison with the positive control), and/or simplified the presentation of these results (e.g. with Table 3 and Figure 1)?
Thank you for the feedback. The statistical significance of each of the results was in fact obtained in comparison to both the positive control as well as being normalized against the scrambled siRNA controls. In Table 3, all shaded blocks except for those containing an asterisk were statistically significant after being normalized against the respective scrambled siRNA controls. For Figure 1, the † indicated the statistical significance of results normalized against the P91C scrambled control as this was the only control that showed statistically significant differences to the positive control. This has been clarified further in the figure legend as per both your, and the other reviewer’s, comments and suggestions.
I suggest to modify, throughout the document, the use of “post-exposure treatment” (e.g. line 214) or “not typically representative of the average exposure” (lines 360-361), which I do not consider suitable for in vitro studies. Simplify if with “post infection” for example.
Thank you for this suggestion. Line 214 and other lines where this appears has been modified with “post-infection trials”, as suggested by the reviewer.
For line 360, we agree that the use of these words is not applicable for in vitro studies, but in this case its usage is in reference to developing a treatment option that would be applicable to relevant real-life situations. To clarify this potential ambiguity in the text, we have made further clarifications. The sentence in question now reads: “As a two-hour delay between viral challenge and treatment is not typically representative of the average exposure in a real-life setting (i.e. bite exposure treatment-seeking behaviour)…”
In the discussion section, I suggest also to the authors to discuss more about the potential limitation of this approach in terms of spectrum of antiviral activity. Indeed, siRNAs and aptamers were designed against the CVS strain. What will be the effect on other and most divergent RABV, even other lyssaviruses? Do the authors select conserved regions in the N and P proteins?
The authors did target conserved and immunologically important regions of both the N and P gene to attempt to address this potential limitation, as mentioned in the Introduction and methods as follows: “siRNAs targeting the RABV N and P gene were developed due to their roles in RABV replication, their high copy numbers (N gene due to transcriptional bias [52]) and their exemplary role in immune suppression and evasion. Aptamers were developed using cell-selection targeting RABV-infected cells as opposed to specific gene products, enabling the recognition of a diverse range of antigens and cell morphologies that present upon infection by RABV.”
However, to further clarify the reviewer’s concerns, the authors have included this point in the limitation section of the discussion that has been added in based on the reviewer’s comments and suggestions. We have addressed this specific point with the following sentence: “further investigation into the ability of these siRNAs and XAs to target other strains and variants of RABV would be required if these are to be applied as a rabies therapeutic globally.”
Similarly, I did not see any discussion about the viral inhibition effects of the negative siRNA controls. Is it common in such approach? What about the absence of negative controls with the aptamers and the mixed assays?
Thank you for the suggestion. We have addressed this concern in the new limitations section that has been added to the manuscript and briefly discuss a potential reason for this. The additional text now includes the following: “The non-specific effects of the scrambled siRNA would warrant further investigation, as based on the statistical results, the addition of certain short RNA sequences elucidated some viral inhibition, despite not being target specific. This viral inhibition could be due to the stimulation of an innate immune response through the binding of the short RNA with toll-like receptors (Marques & Williams, 2005).”
The absence of negative controls for the aptamers has been addressed in a previous comment.
Minor comments
- Line 11: Develop “siRNA”.
Thank you for your suggestion. This has been amended and written out in full.
- Line 21: Develop “RIG”.
Thank you. This has been amended and written out in full.
- Line 63: If I am not mistaken, this timeframe is not anymore indicated in the 3rd WHO report, replace by “People with WHO category II or III exposures should receive PEP without delay as an emergency procedure”. The same with the WER 2018 (No 16, 2018, 93, 201–220) with “promptly started after an exposure”. I suggest to modify accordingly.
Thank you for highlighting this older reference. You are correct and the text has been modified accordingly as follows: “Rabies PEP is only effective before the onset of clinical symptoms and should be initiated without delay.”
- Lines 164-165: How many cells were used per well?
At the time of challenge with CVS-11, there were approximately 1,68x105 cells per well. This has been added into the text in this section.
- Line 216 (and for all the manuscript): Replace “pMol” with “pmol” (quantity) or “pM” (concentration).
Thank you, this has been rectified throughout.
- Figure 1, 2, 3 and 4: Indicate in the legend indicate what the intervals refer to.
Thank you for this suggestion. The figure legends have been improved and reflect the the y-axis interval are the mean percent reduction.
- Figure 3: Asterisks and numbers are superimposed, which make the reading difficult.
Apologies. It seems that the asterisks had shifted and therefore were not clear. All of the figures have been reviewed and remade where necessary to ensure that the asterisks and numbers (and the figures overall) remain clear.

Round 2
Reviewer 2 Report
General comment
I appreciate the work done by the authors regarding the different comments of the reviewers. The manuscript has been clearly improved, and it reads easily. The authors have also responded to almost all of my comments and I thank them for that.
However, I have still two major comments and a limited number of minor comments.
Major comments
Line 446- 448: My main concern about XA remains on the specificity of these aptamers. The authors indicated, in these lines, that the fact of targeting RABV-infected cells rather than specific gene targets, make them potentially more diverse and applicable to various RABV strains and other RABV variants. But how can they be sure that these XA target viral compounds rather than cellular compounds which will affect the infectivity process? Indeed, I rapidly look to the AM Biotechnologies X-Aptamer Selection Kit protocol and did not find any counter-selection, such as against non-infected cell in the current study step (which is not the case in classical SELEX process for example). So how can we be sure that the effective XA are not directed and specific only to cellular targets of MNA cells, and not to other cell lines or even other cell species? In this case, it could limit the interest of their applications. Application to other cell lines could be a way to answer to this question.
Another comment relies on the reply of the authors to the viral inhibition effects of the negative siRNA controls, which is informative. However, will it be a possibility that the authors can try some limited experiments in this way, without engaging in long and tedious manipulation but by looking by RT-qPCR some target genes of these pathways on negative cells for example?
Minor comments
- Line 14: Replace “lyssavirus” with “rabies”.
- Line 15: Add “(pi)” after “post-infection”.
- Line 76: Replace “Rabies lyssavirus” with “rabies”, as you refer to the virus rather than the species.
- Line 88 : Add a comma after “activated”.
- Lines 490-491: incomplete sentence.
- Line 550: Correct the typo “d”.
- Figure 3: Indicate the significant values (with an asterisk or other) on this figure. In this figure, the chimera24 of UPRET 2.01-N8 was really statistically different from XA + si RNA mixture?
Author Response
Reviewer 2: Round 2
Author’s response:
We would like to thank the reviewer for accepting to revisit the revised form of the manuscript and for continued inputs towards its improvement. We have considered all comments to be fair and responded with the necessary changes as best we could.
Major Comments
Line 446- 448: My main concern about XA remains on the specificity of these aptamers. The authors indicated, in these lines, that the fact of targeting RABV-infected cells rather than specific gene targets, make them potentially more diverse and applicable to various RABV strains and other RABV variants. But how can they be sure that these XA target viral compounds rather than cellular compounds which will affect the infectivity process? Indeed, I rapidly look to the AM Biotechnologies X-Aptamer Selection Kit protocol and did not find any counter-selection, such as against non-infected cell in the current study step (which is not the case in classical SELEX process for example). So how can we be sure that the effective XA are not directed and specific only to cellular targets of MNA cells, and not to other cell lines or even other cell species? In this case, it could limit the interest of their applications. Application to other cell lines could be a way to answer to this question.
We understand this concern as it relates to non-target specificity and potentially limited application, should this prove to be the case. However, this was considered in the experimental design and a negative selection step was indeed included. This was the first step in the X-aptamer selection protocol (Cellular V16.2), during which the authors collaborated with AM-Biotechnologies to revise. To clarify this in the manuscript, we have included a sentence in the methods that informs readers of the negative selection step as follows: “The selection method included positive-selection steps and a negative-selection step to reduce potential non-specific binding of the XAs to uninfected cells.”
In terms of your concern regarding specificity to MNA cells only, this is indeed a valid point that would need to be investigated further, depending on the targeting of the XAs themselves. As such, this issue is addressed in the discussion section (limitations) where the need for additional research into the binding targets and for more extensive in vitro and in vivo trials are stressed.
Another comment relies on the reply of the authors to the viral inhibition effects of the negative siRNA controls, which is informative. However, will it be a possibility that the authors can try some limited experiments in this way, without engaging in long and tedious manipulation but by looking by RT-qPCR some target genes of these pathways on negative cells for example?
This point is indeed important to investigate as a means to better understand the impacts of the siRNA alone on negative cells and their potential interaction with cellular and immune pathways. We have highlighted this as follows: “The potential therapeutics developed in this study could not only be further refined beyond proof-of-principle but should subsequently be tested in different cells and animal models to better understand the interactions with immune responses and the overall efficacy of the molecules.” Unfortunately, at this time we are unable to perform additional experiments in terms of the scope of this manuscript.
In addition, the point that this study only investigated MNA cells, bears relevance. While MNA cells do have some immune response capabilities, this would certainly not be reflective of the potential implications on a live host, meaning that either in vivo experiments would be required, or that multiple representative cell lines would be required. Such additional experiments should be undertaken in a comprehensive manner to address all of these questions, based on the proof of principle that we have demonstrated here. These limitations and the need for additional investigation is discussed in the Discussion section.
Minor comments
- Line 14: Replace “lyssavirus” with “rabies”.
Thank you. This has been changed accordingly.
- Line 15: Add “(pi)” after “post-infection”.
Thank you. This has been changed accordingly.
- Line 76: Replace “Rabies lyssavirus” with “rabies”, as you refer to the virus rather than the species.
Thank you. This has been changed accordingly.
- Line 88: Add a comma after “activated”.
Thank you. This has been changed accordingly.
- Lines 490-491: incomplete sentence.
Apologies, this was a remnant from the modification of the document and was missed in the final review before re-submitting. This has been removed.
- Line 550: Correct the typo “d”.
Thank you. This typo has been removed.
- Figure 3: Indicate the significant values (with an asterisk or other) on this figure. In this figure, the chimera24 of UPRET 2.01-N8 was really statistically different from XA + si RNA mixture?
The statistically significant differences observed between chimera24 and the other experimental combinations was based on the Tukey test (ANOVA with post-hoc Tukey Honestly Significant Difference (HSD) test) which relies on the mean values of each of the experimental combinations. Therefore, the mean of the results from the chimera24 resulted in that test overall being statistically significantly less effective to the other approaches. While this may potentially be an obvious determination, the results from that test highlighted that the other combinations were not overall significantly different to one another. As this is based on the mean results, we believe it would be misleading to place asterisks on the graph to demonstrate significant difference and therefore have left the figure itself unchanged. However, to clarify this point and avoid confusion, we have altered the figure legend to include the phrase “mean of the” as follows: “The mean of the results from the Chimera24 trials were statistically significantly…”

Round 3
Reviewer 2 Report
I thank the authors for their reply, which I consider satisfactory regarding my last comments.